
# Debris Flow event on Osorno volcano, Chile, during summer 2017: New interpretations for chain processes in the Southern Andes

Ivo Janos Fustos-Toribio[1], Bastian Morales-Vargas[2,3], Marcelo Somos-Valenzuela[3,4*], Pablo Moreno-Yaeger [1,5], Ramiro Muñoz-Ramirez[2], Ines Rodriguez Araneda[2], Ningsheng Chen[6]

[1] Department of Civil Engineering, University of La Frontera, Francisco Salazar 1145, Temuco, Chile
[2] Departamento de Obras Civiles y Geología, Facultad de Ingeniería, Universidad Católica de Temuco, Rudecindo Ortega 02950, Temuco, Chile
[3] Department of Forest Sciences, Faculty of Agriculture and Forest Sciences, Universidad de La Frontera, Av. Francisco Salazar 01145, Temuco, Chile, 4780000
[4] Butamallín Research Center for Global Change, University of La Frontera, Av. Francisco Salazar 01145, Temuco, Chile, 4780000
[5] Department of Geoscience, University of Wisconsin–Madison, 1215 West Dayton St., Madison, WI 53706, USA.
[6] Key Laboratory of Mountain Hazards and Surface Processes, Institute of Mountain Hazards and Environment, Chinese Academy of Sciences, Chengdu 610041, China

Correspondence to: Marcelo Somos (marcelo.somos@ufrontera.cl)

Abstract. Debris flow generation on volcanic zones at the Southern Andes has not widely studied, despite the enormous economic and infrastructure damage that these events can generate. The present work contributes to the understanding of these dynamics based on a study of the 2017 Petrohué debris flow event from two complementary points of view. First, a comprehensive field survey allowed to delimitate that a rockfall initiated the debris-flow due to intense rainfall event. The rockfall lithology corresponds to lava blocks and autobrecciated lavas, predominantly over 1500 m.a.s.l. Second, the process was numerically modeled and constrained by in situ data collection and geomorphological mapping. The event was studied by back analysis using the height of flow measured in road CH-255 with errors of 5%. Debris flow volume has a high sensitivity with the initial water content in the block fall zone, ranging between $4.7x10^5$ up to $5.5x10^5$ m$^3$, depending on the digital elevation model (DEM) used. Therefore, debris flow showed that the zone is controlled by the initial water content available previous to the block fall. Moreover, our field data suggest that future debris flows events can take place removing material from the volcanic edifice. We conclude that similar events could occur in the future and that it is necessary to increase the mapping of zones with autobrecciated lava close to the volcano summit. Finally, the study contributes to understanding debris flows in the Southern Andes since the Osorno volcano shares similar features with other stratovolcanoes in the region.





## 1 Introduction

Landslides processes are among the most important natural hazards in developing countries due to their low resilience,
generating damage to human life, property, and engineering projects in all the mountainous areas of the world every year
(Martha et al., 2010; Alimohammadlou, 2013; Sepúlveda et al., 2014; Fustos et al., 2020). Debris flows are an important
type of mass wasting, described as one of the most dangerous of these processes due to their high velocity, the damage that
they cause, and the extensive areas affected (Jakob & Hungr, 2005). Nevertheless, debris flows in volcanic zones have not
been evaluated in detail, where only primary volcanic-originated processes like lahars have been studied. The present work
evaluates the generation of debris flows, taking the 2017 Petrohué event as a case study. This event caused severe economic
losses to one of the most popular tourist attractions in southern Chile (INE, 2018).

Debris flows are very destructive processes in active zone in the Andes, especially in volcanic areas independent of their
trigger (Sosio et al., 2011). The northern Andes shows examples like the 1985 Nevados del Ruiz eruption. The volcanic
activity triggered a lahar flow, which claimed at least 25,000 lives (Naranjo et al., 1986). In December 1999, a mud and
debris flow in Venezuela caused the loss of 30,000 lives (Wieczorek et al., 2000). Rock and soil movements, debris
avalanches, debris, and mudflows, and the resulting floods destroyed about 40 km of the Trans-Ecuadorian oil pipeline and
the only highway from Quito to Ecuador's northeastern rain forests and oil fields. This phenomenon was caused by heavy
rain and two earthquakes in 1987 (Schuster et al., 1996). In 2017, a rainfall-induced landslide event with more than 600
shallow landslides was triggered in Colombia. Following the intense rainfall, landslides and subsequent Mocoa Debris Flow
(MDF) event killed to 333 people (García-Delgado et al., 2019). Moreover, the Central Andes has experienced massive
debris flow events like the ones in the Lastarria volcano (Rodríguez et al., 2020). The collapse of part of the edifice triggered
a 270 km/hr volcanic debris avalanche. Catastrophic debris flows occurred on Huascarán (an extinct volcano), Peru, in 1962
and 1970, triggered by ice and rock, which were swept down from the north peak of the mountain because of an earthquake
(Mw 7.9). The 1962 and 1970 events are estimated to have caused ~7,000 deaths (Evans et al., 2009,; Tacconi et al., 2017;
Bueechi et al., 2018). Finally, only scarce volcanic related debris flows have been reported in the Southern Andes. An
hyperconcentrated flow occurred days after the 2008 volcanic eruption at Chaitén volcano. The trigger of the event is related
to intense rainfall over ash deposits within days of cessation of the eruption (Pierson et al., 2013). Therefore, the connection
between debris flow and volcanic environments in the Andes has been covered in the past in Northern and Central Andes.
However, they are still only superficially studied in the Southern Andes.

In the Southern Andes, volcanic edifices are covered by materials that could produce recurrent debris flows, molding the
relief. The debris flows reported currently in the literature on volcanoes are mainly the result of snow and ice melting during
eruptions in lahar form (Johnson and Palma, 2015; Major et al., 2016; Thouret et al., 2020). Nevertheless, few works
addresses the relation between the debris flows generated by soil conditions in volcanic systems and the morphology of the
edifice. Their spatial and temporal extension has also been little studied; these are very important since the number of debris
flows in volcanic systems has increased in recent years (Pierson, 1995; Aguilar et al., 2014; Thouret et al., 2020).





The present work seeks to advance our comprehension of the generation of debris flows in volcanic edifices in the Southern Andes. An atypical debris flow event of January 8th, 2017 (southern summer) on Osorno volcano (Southern Andes) was assessed. The geomorphological and geological factors influencing the generation of the debris flow event were studied, estimating the conditions which triggered the event through back analysis using the r.avaflow model. Finally, we discuss

whether the event might be recurrent in time or is part of the normal cycle of the volcano; this knowledge will assist in assessing the risk of debris flows in the Southern Andes.

## 2    Study area

Osorno is a stratovolcano that is part of the active volcanic arc of the southern Andes, called the Southern Volcanic Zone (SVZ). The SVZ is a continuous volcanic arc of 1400 km long, extending from 33.3º to 46º S (Stern, 2007; Moreno &

Gibbons, 2007).  The study area is the south-eastern flank of Osorno volcano (41.1054ºS, 72.4961ºW), beside Route CH-255 (Figure 1), which connects the villages of Ensenada (on the eastern side of the Lake Llanquihue) and Petrohué (on Lake Todos Los Santos).

The area has a total population of 44,578 people with a density of 11.0 per km2. The topography makes CH-255 routethe only connection between Petrohué and Ensenada, and regular aero transport is impossible except for a few flights by

helicopters, unavailable for the local population. Therefore, CH-255 route becomes a critical infrastructure for local development. This had led to call that route as "El Solitario Pass" (Lonely Pass) Finally, note that the village of Petrohué has a population of 193 people, surpassing 3,000 in summer. The village does not have the capacity for autonomous subsistence, depending on the food and services of Ensenada.

### 2.1    Geological setting

Osorno volcano is a mainly basaltic Pleistocene to Holocene composite volcano (Figure 1). Its part of an SW-NE volcanic alligment along with  three other volcanoes: La Picada, Puntiagudo, and Cordón Cenizos (Moreno et al., 2010), oriented obliquely to the main volcanic arc and the Liquiñe-Ofqui Fault System (LOFS). The chain orientation suggests that the area is an active transtensional zone of the crust, with mafic magma extrusion during the Quaternary (Cembrano & Lara, 2009; Moreno et al., 2010). The surrounding area shows pre-LGM (Last Glacial Maximum) Pleistocene volcanic rocks including

tuffs, breccias, and lava originating from the northern zone of the volcano (Moreno et al., 1985).

Alluvial fans in Osorno volcano are composed of unselected sandy rich matrix polymictic gravels, organized in meters thick banks. These form the current filling of the gullies on Osorno volcano, together with alluvial deposits generated by the re-working of moraine deposits and old lahar fans. Debris flows have also been recorded, triggered by snow-melt and intense rainfall in the zone (Moreno et al., 2010). Alluvial deposits exist associated with debris flows and rockfalls in the zone. The

granulometry ranges from sand to gravel, and the deposits extend to the shore of the Lake Todos Los Santos (Garrido, 2015).


## 2.2   Records of rainfall-induced landslides

The zone has suffered recurrent debris flow events during periods of variable precipitation, so the factors which triggered these events are strictly unknown (Garrido et al., 2017; Garrido et al., 2018). For example, debris flows and mudflows occurred on June 2nd, 2015, associated with a front of intense and prolonged precipitation over CH-255 (El Solitario pass).

These flows damaged five houses and four barns, as well as destroying one water tank and some pipework of the second water tank of the drinking water supply network of the village of Petrohué (Garrido, 2015). On January 8th, 2017, large debris flows occurred again on the eastern sector of the volcano's south flank, during a front of intense precipitation. 94 mm of precipitation in a period of 24 hr were recorded in Ensenada, with the 0° isotherm above 3,000 m.a.s.l. The Petrohué debris flow was an atypical event in that it occurred outside the season of intense precipitations; it caused serious material

and economic damage to one of the most popular tourist attractions in southern Chile. The road was blocked in 5 different places, and tourists were cut off for several hours in the area of the Saltos de Petrohué (Garrido et al., 2017; Garrido et al., 2018).

Figure 1 Geological map of Osorno volcano based on Moreno et al. (2010).

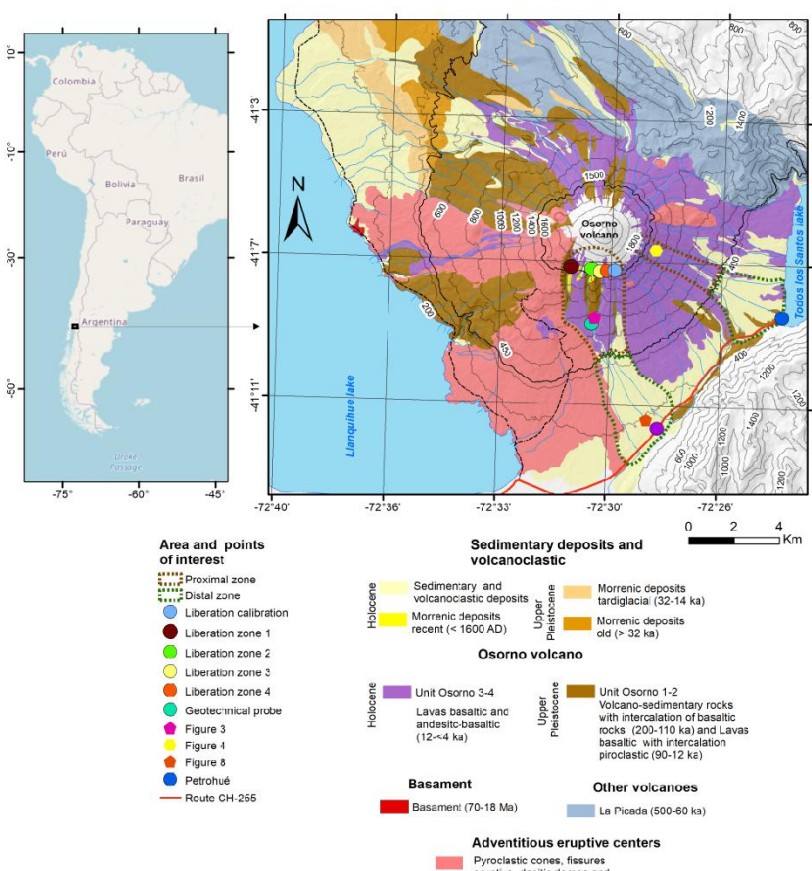





## 3    Methodology

The 2017 Petrohué event was studied to understand the factors which control the occurrence of debris flows in the Osorno volcano. The geomorphological factors influencing debris flow generation were studied by collecting data in the field and defining the release zones. The field results were used as border conditions in numerical modelling of the event by back analysis, taking flow heights recorded in technical reports and photographs to define zones of comparison (Figure 2).

<p style="text-align:center">Figure 2 Short methodology.</p>

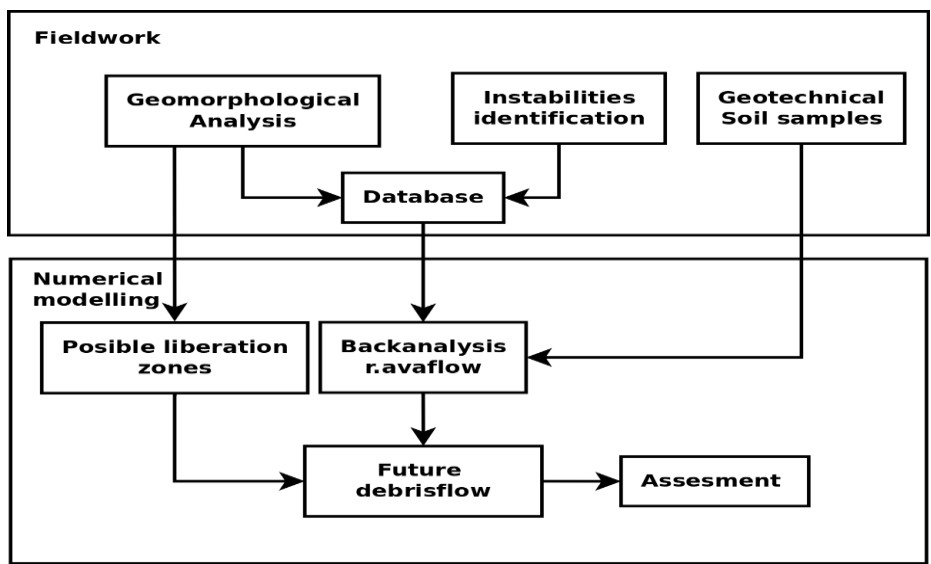

### 3.1    Field evicence

The mechanisms which generated the 2017 Petrohué debris flow event were studied in the area, approaching by the "El Solitario" pass (41.1943ºS, 72.4759ºW). Areas affected by the flow were identified and mapped in detail. In-situ debris flow deposits analysis is critical to establish a rheological model to be used. Specifically, the grain size distribution transported was assessed, indicating if the flow corresponds to a hyper-concentrated process or not. Hence, geomorphological characteristics were analyzed, noting particularly slopes susceptible to movement in intense precipitation events. The geomorphological features were evaluated by measuring slopes and heights with metric rules and qualitative analysis of depositional structures related to the debris flow.

Due to the extensiveness of the area, elevation and cannel gradient data is derived from the two different DEMs, SRTM and ASTER. Additional information regarding distance measurements such as side slopes, channel depth and the maximum width of the landslide was evaluated in the field using metric rulers. Specifically, the width of the landslide is identified and georeferenced using a handle GPS to constrain the numerical modelling results. On the CH-255 road, the final height of debris flows is established as 1.5 m from Garrido et al. (2017). Different soil deposits allowed understood the mobilization features, checking if it corresponds to a non-Newtonian flow. Liberation zones close to the Volcano summit, were physical





weathering of the lava flow deposits exposed by pluvial erosion was assessed in the field. Scarps with potential rockfalls of unstable blocks were identified, measured, and georeferenced to stablish these zones in the follow numerical model. Finally, debris flow runout was estimated measuring the channel distance between the liberation zone and the CH-255 limit using navigation GPS.

Geomechanical properties of the mobilized material were characterized using geotechnical testing. Three unaltered geotechnical soil samples were extracted for a direct shear test (Figure 1). The samples were extracted from a depth of 40 cm (41.1843ºS , 72.4652ºW) to avoid integrating possible organic material in each sample. Because the soil corresponds to a granular soil, we used a split cube (Table  1). The sampling probe was removed from unsaturated media without water table presence. The sample water content was determined by drying at a constant temperature of 60° C (ASTM, 2019). The

geotechnical shear test was carried out deforming a specimen at a constant controlled stress rate throughout the period of deformation based on ASTM D3080 / D3080M-11 standard (ASTM D3080, 2020). The estimated values of the cohesion and internal friction angle was integrated into a database to be used during the back analysis phase.

Table  1 Properties obtained by direct shear test. Geotechnical results incorporated into the r.avaflow model as constraints.

| Physical properties per test sample | | | |
|---|---|---|---|
| Physical properties | Test sample 1 | Test sample 2 | Test sample 3 |
| Water content (%) | 3.31 | 3.30 | 3.27 |
| Natural unit weight ($g/cm^3$) | 1.62 | 1.56 | 1.63 |
| Dry unit weight ($g/cm^3$) | 1.57 | 1.51 | 1.58 |
| Unit weight after consolidation ($g/cm^3$) | 1.67 | 1.60 | 1.64 |
| Lateral displacement rate used (mm/mi  ) | 0.50 | 0.50 | 0.50 |
| Lateral displacement achieved (%) | 10.00 | 10.00 | 10.00 |
| Maximum shear force ($kg/cm^2$) | 0.73 | 0.51 | 0.34 |
| Normal force ($kg/cm^2$) | 0.80 | 0.40 | 0.20 |

## 3.2    Debris flow modelling

Representing debris flows is currently a challenge due to the possible changes of phase, which may occur during the process from generation to stabilization. Therefore, the r.avaflow model was used to evaluate the fall of blocks on saturated/unsaturated zones and their subsequent evolution as non-Newtonian flow (Mergili et al., 2017; Mergili et al., 2018b). Measurements of soil water content into the fall of blocks are not available. Therefore, various water content scenarios were carried out.

### 3.2.1    Back-analysis

The r.avaflow model was applied by back analysis, taking as constraints the cohesion data and the angle of internal friction obtained from the undisturbed soil samples collected in the field (Figure 1). The back-analysis considered a final height of





1.5 m in route CH255, according to reports of SERNAGEOMIN. Moreover, volumes of liberation zone were integrated from the evidence collected in the field. The model used a two-phase parametrization based on Pudasaini (2012). First, the solid

phase corresponds to the lava and auto-breccia fall; meanwhile, the second phase is associated with the debris flow generation under saturated/non-saturated water conditions. During the solid phased, a mohr -coulomb plasticity approach allowed it to be estimated for the stress. The fluid stress was modeled as a solid -volume -fraction -gradient -enhanced non - Newtonian viscous stress (Pudasaini, 2012; Mergili et al., 2017). Let $u_s = (u_s, v_s, w_s)$, $u_f = (u_f, v_f, w_f)$ and $\alpha_f$, $\alpha_f = (1 - \alpha_s)$ denote the velocities, and volume fractions for the solid and the fluid constituents, denoted by the suffix s and f, respectively.

If we take the $\eta_f$ correspond to the fluid viscosity with isotropic stress distribution. The phase-averaged viscous-fluid stresses are modelled using non-Newtonian fluid rheology:

$$\tau_f = \eta_f \left[ \nabla \, \underline{u_f} + \left( \nabla \, \underline{u_f} \right)^t \right] - \eta_f \frac{A(\alpha_f)}{\alpha_f} \left[ (\nabla \alpha_s) \left( \underline{u_f} - \underline{u_s} \right) + (u_f - u_s)(\nabla \alpha_s) \right] \tag{1}$$

The back analysis used a generalized interfacial momentum transfer that included viscous drag, buoyancy, and virtual mass.

Non-Newtonian viscous stress was based on Pudasaini and Mergili (2019), separating the solid- and fine-solid- volume fraction gradients, enhancing the apparent viscous stress estimation in the fluid phase. Parameters not measured in the field were established from similar works and data from the area of Villa Santa Lucía (Somos-Valenzuela et al., 2020; Table 2). In a first approach, parameters presenting serious discrepancies were restricted to the range proposed by Pudasaini (2012) for the simulation of debris flows.

Surface features such as slope are important input data for debris-flow modelling [Qin et al., 2013]. DEM errors introduce uncertainty in terrain representation leading to a poor estimation of the numerical solutions. Given the uncertainty of the (DEM before the 2017 Petrohué event, two DEMs were used as references to assess the sensitivity of changes in elevation. The SRTM and ASTER-GDEM models were used separately, with a spatial resolution of 30 meters. The possible release zones or areas of origin of these debris flows were established from the geomorphological evidence found in the field. The

results of the back analysis were compared with photographs provided by the Chilean Geology and Mining Survey (SERNAGEOMIN) based on "El Solitario" (~ 41.180ºS, 72.462ºW) to assess the performance of the models with different initial water contents.

Table 2Model parameters usd in r.avaflow.

| Symbol | Parameter | Value | units |
|--------|-----------|-------|-------|
| $\rho_s$ | Solid material density | 2800 | $kg/m^3$ |
| $\rho_f$ | Fluid material density | 1000 | $kg/m^3$ |
| $\phi$ | Internal friction angle * | 32.4 | Degree |
| $\delta$ | Basal friction angle | 6 | Degree |



| $C$ | Virtual mass | 0.5 | - |
|---|---|---|---|
| $U_T$ | Terminal velocity | 1 | $m/s$ |
| $P$ | Parameter for combination of solid- and fluid-like contributions to drag resistance | 0.5 | - |
| $Re_p$ | Particle Reynolds number | 1 | - |
| $J$ | Exponent for drag | 1 | - |
| $N_R$ | Quasi-Reynolds number | 4,5 | - |
| $N_{RA}$ | Mobility number | 3 | - |
| $x$ | Viscous shearing coefficient for fluid | 0 | - |
| $\xi$ | Solid concentration distribution with depth | 0 | - |
| $C_{AD}$ | Ambient drag coefficient | 0,02 | - |
| $C_E$ | Entrainment coefficient | -6,69 | $kg^{-1}$ |
| $C_{FF}$ | Fluid friction coefficient | 0,001 | |

3.2.2    Sensitivity analysis

A systematic study has been carried out to represent the debris flow. Mergili et al. (2018a) established that parameters with high sensitivity correspond to the basal friction angle, fluid friction coefficient, and environmental drag coefficient. We used reference values previously considered in the zone (Somos-Valenzuela et al., 2020). Moreover, geotechnical laboratory tests allowed to represent the friction angle and cohesion values adequately. Sources of uncertainty were attributed to surface

representation and initial water content in the head of block fall.  A wide range of initial water content was considered, constrained by the geomorphological evidence at the site, to calibrate the model to observations in the field. Since the initial proportion of water when the hyper-concentrated flow was generated is unknown, we assumed a water content in the initial volume between 40% up to 70%, considering the high porosity of the material involved. Hence, we calibrate the flow runout taking control points of the height of the flow measured on the main road minutes after the event. We also assessed the

quality of the simulations using possible release volumes based on field evidence.





### 3.2.3    Projections

Finally, possible scenarios evaluating the impact of new debris flows in the area were analized. Therein, we defined new unstable release zones identified visually in the field. We identify areas with intense rain erosion, hanging blocks, or fractured rocks. This enabled us to estimate the potential volume transported, and thus to understand the impact of different
debris flows generated in zones which were very close together, but with release at different altitudes.

## 4    Results

The conditions that generate debris flows were evaluated in an active volcanic zone, with reference to the 2017 Petrohué event. Information collected in the field was assessed and compared with numerical modelling using back analysis with r.avaflow.

### 4.1    Field evidence

The debris flow in the distal zone is characterized by poorly sorted volcanic material. The deposited material is supported by a medium-coarse sand matrix (2 mm) along with >1 meter diameter blocks. The mode of the clasts in the sandy matrix varies between 3 to 10 cm, being consistent with previous results (Garrido et al., 2017). Moreover, lateral erosion after the debris flow shows a highly energetic process. Geotechnical measurements showed low unit weight (Table 2) supporting that the
thickness of the flow is moderated. These which favours a faster movement, increasing lateral erosion according to field results [Shu et al., 2018]. The lahar deposits show granulometric differences, with strong graduation from the fracture zone (proximal zone) to the final deposition zone (distal zone). From 400 masl, volcanic deposits alternate with autobrecciated lava flows (Figure 4). The walls are up to 20 meters high with high stratification and different grain size. A pronounced difference in competence was observed between the volcanic deposits and the autobrecciated lava flows, which facilitates
differential erosion and planes of weakness (Figure 3).
Field evidence showed that debris flows are generated by the fracture of basaltic lava over volcanic deposits in the high altitude zone of the study area. Rockfalls occurring above 1500 m.a.s.l. were identified (Figure 1 and Figure 3). This zone presents numerous scarps with pronounced slopes overhanging fluvial drain channels (Figure 4). The remains of the debris flow identified in the field are associated with transported blocks of basaltic lava and primary lahar deposits.
The incision is favored by the presence of very thick lahar deposits (Figure 4), which facilitate the removal and contribution of material to the main channel. A sequence of lahar deposits was observed, overlain by lava flows in blocks up to 1.5 m thick. These occur in regular sequences, leaving alternate levels of erosion and hanging blocks, facilitating the collapse of the lava levels, and generating rockfalls.  The material is characterized by lava with base autobrecciation, no more than 1 m thick (Figure 4). The autobrecciated zone is also heavily weathered (Figure 3); as a result, it can be easily removed, exposing




the center of the lava flow. The lava flow forms a hanging block that can easily fracture and break off. There is evidence of broken-off blocks associated with the central part of the lava flow, due to the instability of the base autobrecciation, with dimensions of up to 2 m. The blocks fracture continuously in the lava runs perpendicular to the slope of the main channel where these debris flows occur. This breaking off of material due to weathering contributes to the main channel, generating powerful debris flows as evidenced by the deposits further down the slope, as a consequence of the continuous rockfalls. The

intermediate zone has narrow drainage channels and an increase in the incision to a depth of around 15 m (Figure 4).

Figure 3 Set of scarps at approximately 1500 masl; the base breccia of the lava flows is heavily eroded.

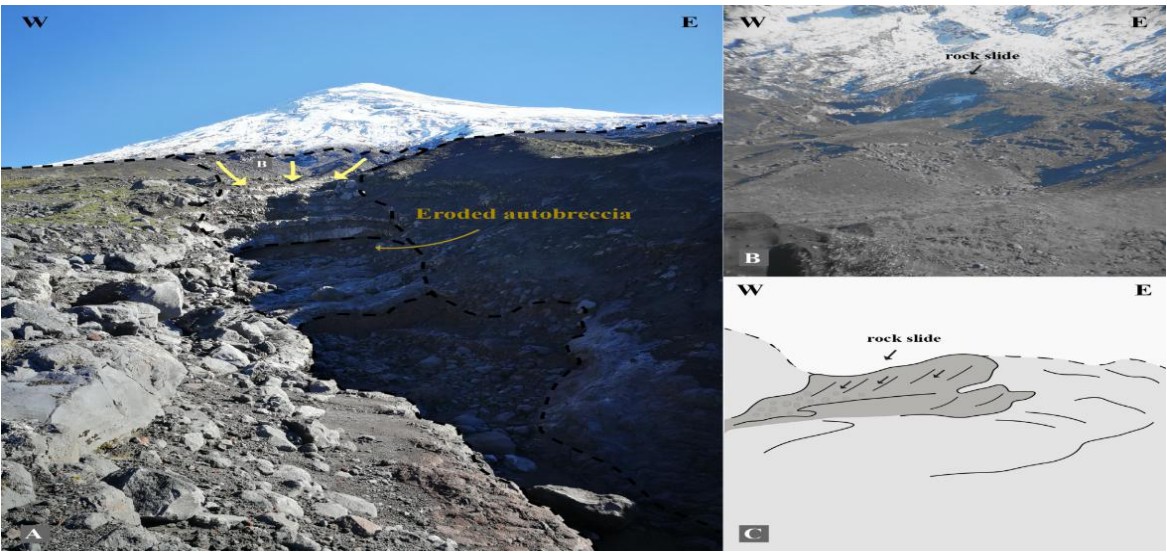



Figure 4 Fifteen-metre high scarp showing the stratigraphy of the volcano at this point, composed of alternating volcanic and lava deposits.

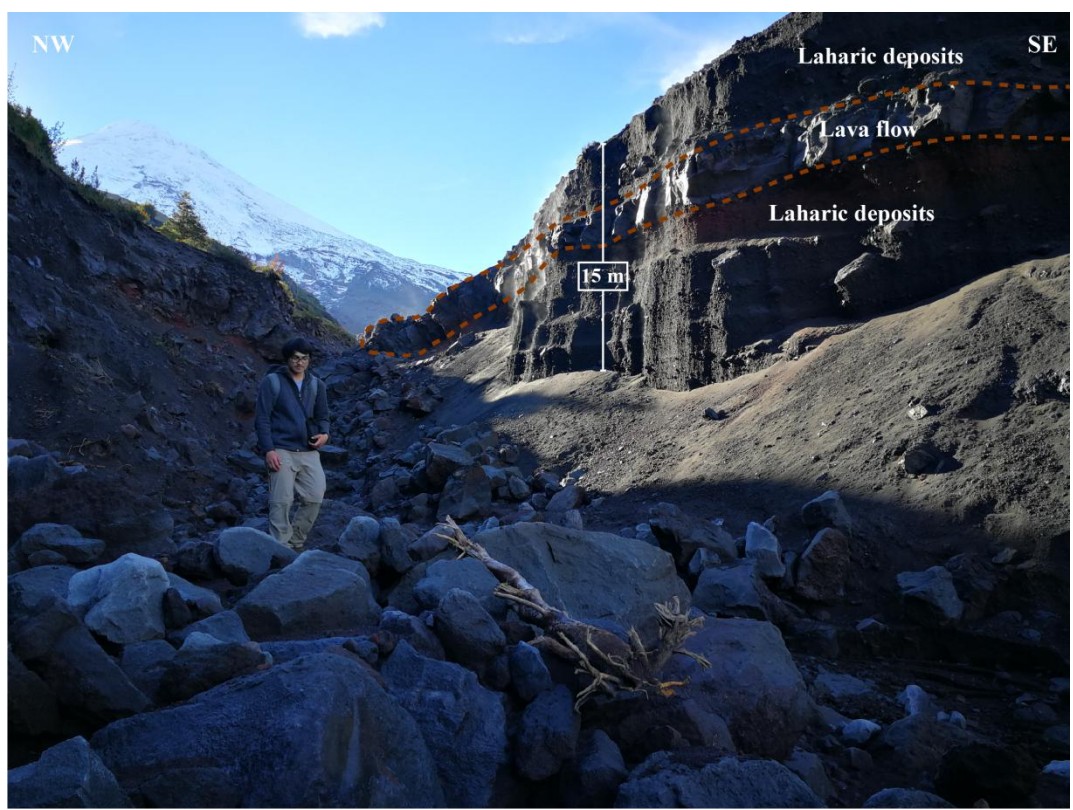


## 4.2    Debris flow modelling

To understand the scope of the runout generated in the release zones of material recognized in the field, the flow was modeled in r.avaflow. Different initial water content into the simulations and DEMs differences allowed understood the uncertainty of the main initial input. The results of the back analysis, restricted by geotechnical soil data (¡Error! No se 235   encuentra el origen de la referencia.), showed that the model that presented the smallest mean error was the simulation using DEM SRTM and 70% water content (error 5%). The results show that in both simulations, the flow covers a large part of Route CH-255, to a 1.59 meters depth with DEM SRTM and 1.58 meters depth with DEM ASTER (Figure 5). Our results indicate that DEM ASTER presented a larger underestimation of the height of -25% with a water content of 65%. With a water content of 40%, DEM ASTER produces an overestimation of 63% against the value measured. These results suggest 240   that the initial quantity of water during the collapse has a large influence on the area affected.

The area affected by the debris flow was estimated at between 7.5 million m² and 7.8 million m²  by DEM ASTER, and between 8.3 million m² and 8.6 million m² by DEM SRTM (Figure 5a). DEM SRTM produced a variation in the flow to the south, increasing the impact in Route CH-255; in contrast, DEM ASTER concentrated a larger part of the material towards


the north (Figure 6). The maximum height was calculated at between 5.55 and 6.74 meters by DEM ASTER, and between
6.33 and 7.69 meters by DEM SRTM. A progressive increase in the maximum height as the percentage of water in the
release zone increased was noted in the simulations. DEM SRTM presented greater heights than DEM ASTER in all the
simulations. The major debris flow height is reached when the initial volume has 70% of the water in both DEMs. Finally,
all the simulations reached populated zones, regardless of the DEM used. The maximum height with DEM SRTM is 1.87
meters with a water content of 40% of the initial water content, while with DEM ASTER, it is 2.13 meters with a water
content of 60%. The greatest heights are concentrated with a water content of 50%-60% of the total material released.

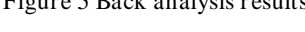

Figure 5 Back analysis results.

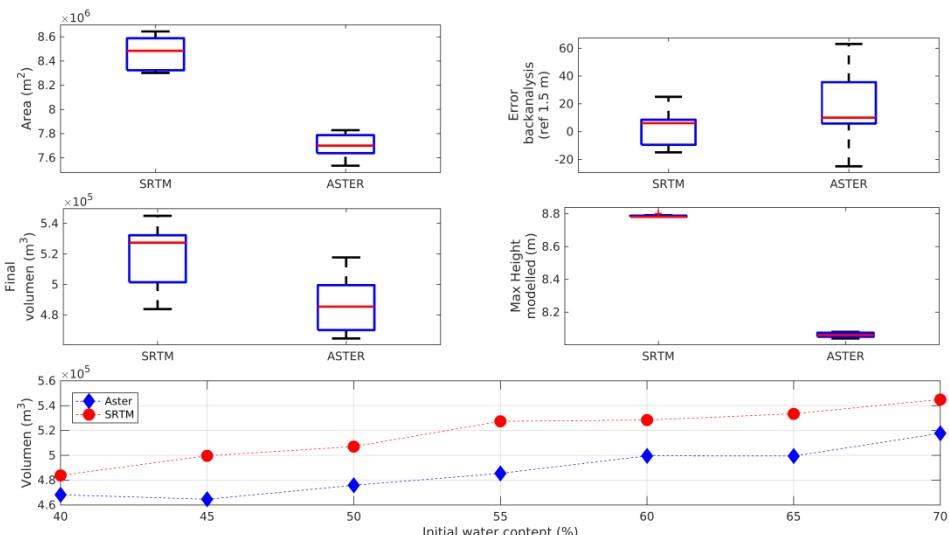




Figure 6 Simulation in El Solitario sector, taking the release zone from the results obtained in the field. Left: Model with 60% water content in DEM ASTER. Right: Model with 60% water content in DEM STRM.

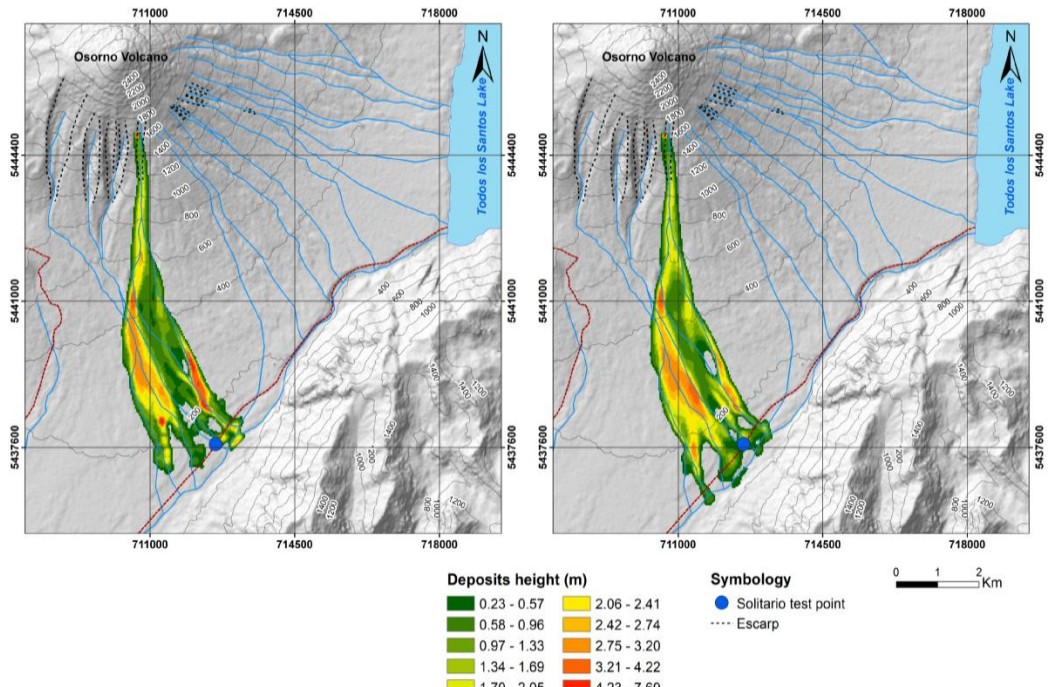


In addition to the zone identified as the source of the Petrohué event of 2017, three zones with unstable autobrecciated lava were cataloged as possible debris flow generation zones (Figure 1 and Figure 3). The information collected in the field showed that the gullies in these zones are severely weakened, so debris flows produced by rockfalls can be expected imminently. Additional release zones to those in r.avaflow were estimated with calibration based on the parameters of the

back analysis model. Our results indicate that the area potentially affected by the debris flow varies between 3.1 million m$^2$ and 3.8 million m$^2$ using DEM ASTER, and between 1.3 million m$^2$ and 4.4 million m$^2$ using DEM SRTM (Figure 7). DEM SRTM produces the largest area affected with 60% water content in the initial volume released and DEM ASTER with 70%.

According to the back analysis, a rockfall with 50% water content is capable of transporting a potential volume of 138,628

m$^3$; the potential with 70% water is 148,830 m$^3$. According to DEM SRTM, the potential volume with 50% water content is 32,039 m$^3$, increasing to 177,399 m$^3$ with 70% water. In this case, the greatest volume is generated when the proportion of water is equal to 70% of the volume initially released, and the highest value is given by DEM SRTM simulation. The lowest value is obtained with a water content of 50%, using DEM SRTM. Finally, the maximum height for DEM SRTM is 0.76 meters, when the water content is 70% of the volume released. In this simulation in DEM ASTER, the debris flow would not

reach Route CH-255 (Figure 7). Our results indicate that the flow released from zone 2 with 60% water content will reach a



minimum height of 1.82 meters in populated zones. Likewise, there are cases in which the debris flow will not necessarily reach the populated zones, suggesting that events of this type are not always recorded.

Figure 7 Impact of different debris flows.

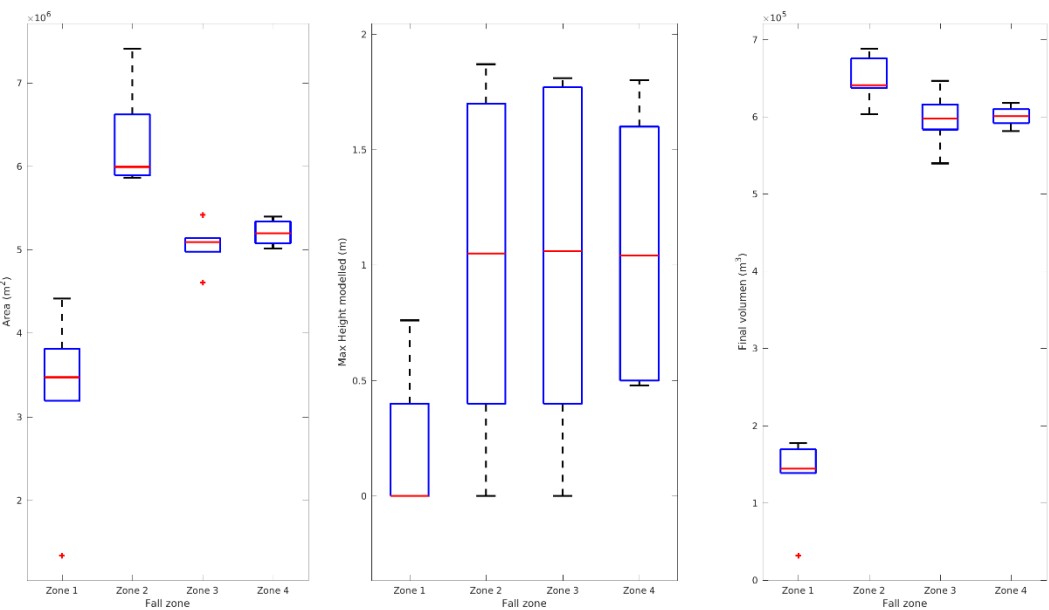

Finally, our results indicate the existence of events that will not generate debris flows even if there is a fall of lava blocks in the channel. This can be seen in zone 1, identified by the evidence collected in the field as an area in which debris flows are generated. On the other hand, a hypothetical scenario of a debris flow generated in zone 2 could lead to debris flows with larger volumes than those observed in previous events (Figure 7). Likewise, it can generate greater flow heights than values recorded to date, leading to more catastrophic events in populated zones. This risk has not been considered to date and needs

to be assessed with care.

5    Analysis and Discussion

5.1    Field evidence

The geomorphology of the Osorno volcano is characterized by the alternation of basaltic lava flows overlain by large volcanic deposits (Figure 4). The fractures identified in the lava flows were probably generated by gravitational effects.

Water can then enter the rock/soil fractures, making them more likely to break off and transporting the material in debris flow form. Conditions are therefore favorable for chain processes culminating in debris flows when the soil is saturated by





high altitude rainfall. Erosion of the deposits exposes the lava flows; material breaks away and is transported by gravity and/or swept down by the force of the water. Furthermore, the autobrecciation of the lava flows increases the instability of the rock faces in the release zones due to the high porosity of the material (Vezzoli et al., 2017; Schaefer et al., 2018).

The evidence collected in the field showed a heterogeneous distribution of lava and slopes close to the debris flow release zones. In this way, the magnitude and force of the landslides processes may be affected by the spatial distribution of volcanic products, principally lava flows. The results show that above 600 masl, there are many exposed lava layers at the base of the principal fluvial channels. A bedrock channel could increase the velocity of the flow in comparison to an alluvial-type channel. This characteristic suggests that the base could act as a sliding surface for the material (Dufresne et al., 2019),

which is very common in stratovolcanoes in Southern Andes. It could be a smooth surface with lower friction, especially under rainy conditions. This could have a critical influence on the velocity and acceleration of the flow from the higher reaches of the edifice. Lavigne & Suwa (2004), Sheridan et al. (2005), and Aaron & Hungr (2016) suggest that the dynamic of debris flows depends upon the friction of the base surface. Having said this, the distribution of the lava and its autobrecciation play an important role in the generation of landslides, rockfalls, and debris flows in the study zone (Figure

300     8).

Figure 8 A. Distal zone in an alluvial deposit in the El Solitario sector. B. Alluvial deposit 1.2 m thick; the base level presents flow structures with fluvial erosion.

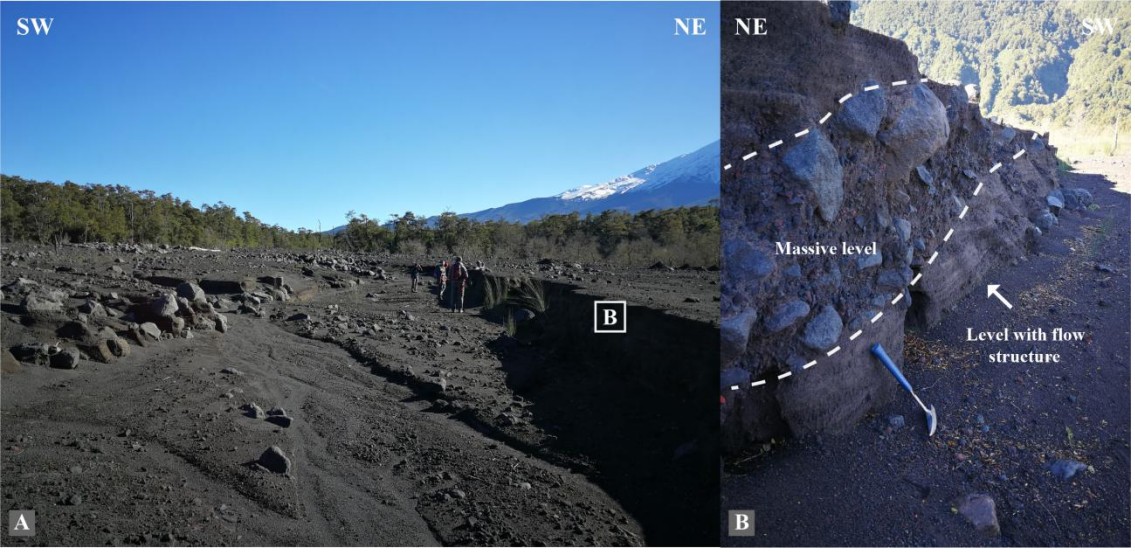

5.2     Debris flow modelling

Our results represented the dynamic of the 2017 Petrohué debris flow with variable errors in the back analysis. Our models were consistent with results obtained in the field, showing a strong influence on the initial water content. The simulations





present high sensitivity to the water content previous to the generation of the event – all the simulations in which the water content in the release zone was higher than 45% reached populated zones. The calibration parameters played an important role in the sensitivity of the numerical model. In the present study, the drag coefficient was established at 0.020, based on Zwinger et al. (2003) and Oyarzún (2019); this value was adopted due to the degree of optimization in the back analysis. A larger or smaller coefficient could produce large deviations in the final height and direction of the flow (Mergili et al., 2018a). The angle of internal friction was determined by a geotechnical study. However, the heterogeneity of the zone could produce substantial changes in φ, so this value must be assessed with great care in future cases. This introduces great uncertainty into the models due to the existence of an imminent bias in the results. It is clear from our results that the use of a DEM introduces a bias in the modelling, which may be directly linked with the methodology by which each product is created and the date of acquisition (e.g., Kääb et al., 2005; Bühler et al. 2011). It must be remembered that processes may occur, which produce changes in the relief that the DEM is unable to capture due to low spatial resolution (Figure 8A and Figure 8B). Alganci et al. (2018) report that SRTM has better vertical accuracy (8.5 m) vs ASTER GDEM (16 m). Vertical uncertainty provides wrong flow routing values, overestimating the final height in some cases. Therefore, DEM product could introduce an additional uncertainty which must be analyzed carefully during calibration and interpretation. Nonetheless, back analysis allows us to estimate variations in the volumes transported and their extreme values, with variation within the same order of magnitude (Figure 6).

The results show that the scope of the debris flow is proportional to the initial water content. Thus the water content available during the collapse of material at the 1500 m contour allows events to occur, which will reach populated zones. Our results show that debris flows are dangerous if the collapse happens with saturation over 50%. We propose that the presence of water in the release zone is explained by local hydrometeorological conditions, i.e., rainfall at high altitude; this is consistent with similar events at Villa Santa Lucía (Garrido et al., 2018; Somos-Valenzuela et al., 2020).

The differences between the initial and final volumes suggest the incorporation of material into the debris flow due to the erosion, which causes movement of the flow (Figure 9). This is supported by evidence in the field, which showed that movable material is available between the distal and proximal zones (Figure 8B). Retreating scarps were observed, which continuously add material to the drainage networks, and this material is available when debris flows occur. Our field results indicate that the largest volume of the material comes from volcanic deposits (Figure 8A).


Figure 9 Conceptual model of chain processes in Osorno Volcano.

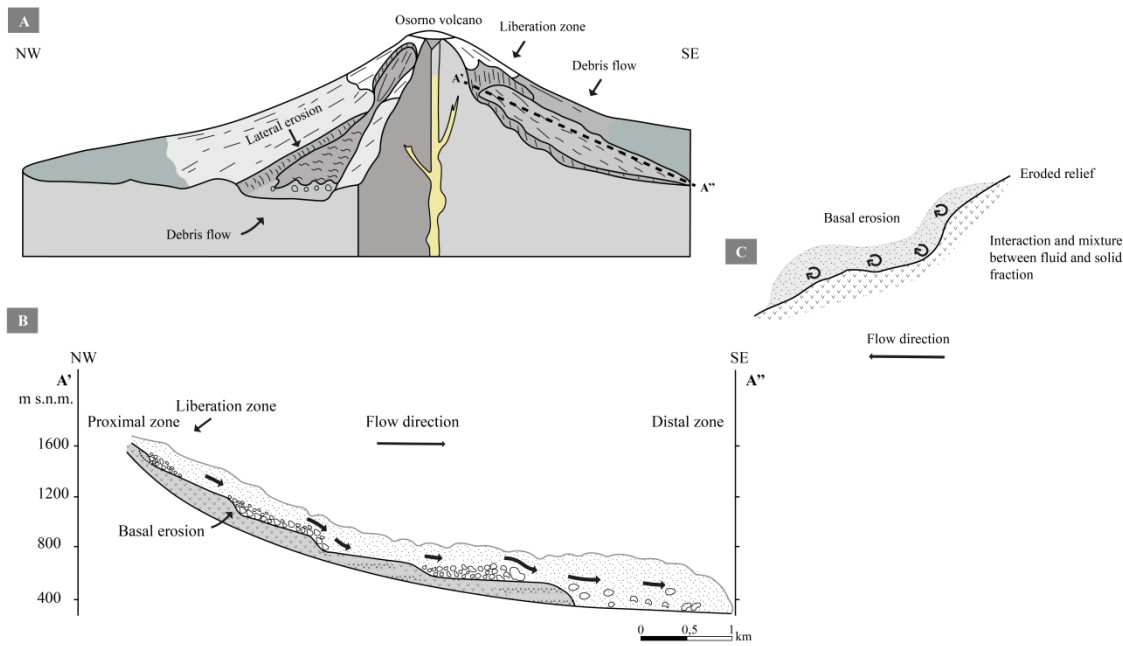


Finally, water-rich mass flows are distinguished by material type, water content, the presence of excess pore pressure, or liquefaction at the source (Calhoun & Clague, 2018). We controlled possible sources of uncertainty such as lithology and topography through a terrain analysis that allowed reducing degrees of freedom. Therefore, the initial water content becomes

an important independent variable due to the unknown value during the main event. The gap of soil moisture stations close to the liberation zone does not allow to constrain the numerical solution with precise detail. Our results show debris-flow generation over 50% of initial water content. The final volume variation of the debris flow varies between 4.8 to 5.6 x 105 $m^3$. The DEM SRTM showed small errors in comparison to ASTER, so, we establish that the volume estimated from SRTM is more suitable for our case. Moreover, SRTM showed more area affected in comparison to ASTER by the debris flow. We

propose that the use of an SRTM scenario correspond to an accurate solution considering lesser error and conservative solutions for future debris flow events. However, we do not establish preliminary conditions for soil liquefaction. To the future, this issue will need to be addressed carefully to improve our understanding of the flow-type landslide in volcanic environments.

Finally, water-rich mass flows are distinguished by material type, water content, the presence of excess pore pressure, or

liquefaction at the source (Calhoun & Clague, 2018). In the present work, it was possible to control sources of uncertainty such as lithology and topography through a terrain analysis that allowed reducing degrees of freedom. In this sense, the component that impacts the most ends up being the initial water content. Our results show debris-flow generation over 50% of initial water content. The final volume variation of the debris flow, based on our calibration results, varies between 4.8 to



$5.6 \times 10^5$ m³. Due to the lower error of the DEM SRTM, we propose that the volume estimated from SRTM is more suitable

for our case. On the other hand, the area affected by the removal is greater with SRTM, for which we propose that the best modelled scenario represents at the same time the worst scenario for the resident population in the study area. However, we do not establish preliminary conditions for soil liquefaction. To the future, this issue will need to be addressed carefully to improve our understanding of the flow-type landslide in volcanic environments.

### 5.3 Future implications

Our projections indicate that the danger to populated areas is strongly dependent on the release zone of debris flows (Figure 7). This suggests that debris flows may repeatedly occur, which are not observed because they are remote from populated areas, increasing the structural destabilization of the volcano in the long term. The release volumes calculated in the present study were defined according to the current stability conditions observed in the field; however, more intense precipitations could lead to more significant rainfall erosion of transportable material (Figure 8B), favoring increasingly violent debris

flows. Such scenarios require the appropriate authorities to propose road design projects as a matter of urgency to evacuate the flows quickly and efficiently. This will improve the mitigation efforts to prevent the population from becoming cut off from the rest of the country. Finally, the geomorphology of the Osorno volcano is not unique in the Southern Andes. For example, Villarrica and Llaima volcanoes show similar conditions to Osorno. We thus have firm grounds for assuming that unexpected debris flows could occur elsewhere in southern Chile, with the difference that these volcanoes are in more

densely populated zones, exposing the population to even more danger.

### 6 Conclusion

The 2017 Petrohué event was studied by back analysis to understand the impact of debris flows occurring on active volcanoes in the southern Andes. Comprehensive in-situ data were used to constrain a numerical model for understanding the debris flow event. For the first time, we evaluate the liberation zone for Osorno Volcano, showing that debris flows occur

due to the collapse of autobrecciated lava flows above 1500 m.a.s.l. (Figure 9A). These results are critical to evaluate future debris flow events in the Southern Andes.

Geomorphological in-situ data determined that the Petrohué event was by a combination of factors such as fluvial erosion and the composition of the volcano, both of which together led to a loss of stability (Figure 9B). The results show an important lithological influence on debris flows in the study zone for the first time. The alternation of volcanic deposits with

lava flows and volcanic deposits with autobbreciated lava flows generate ideal conditions for the continuous development of debris flows during periods when the 0° isotherm is at high altitude. The geomorphological evidenced that block falls and slips occurring mainly above 1500 m a.s.l. becoming into debris flows, increasing the volume transported to the base of the





volcano (Figure 9C). Each event incorporates the material present on the base and sides of the river channels, increasing the final volume. The erosive force associated with the event is conditioned by the amount of water available during the process.

Simulations in r.avaflow showed that the volume varies between 464,564 m³ and 544,903 m³, depending on the amount of water available during the generation phase. Our simulations also indicate that a debris flow from only 45% of water content may reach the populated areas. This suggests that the quantity of water available at the moment of release and along the course of the debris flow is critical. Our results did not allow us to estimate the amount of water available for the 2017 event. However, they do reveal the need to carry out continuous monitoring of the water available in the release zones to determine

thresholds when extreme hydrometeorological events high on the mountain may trigger a debris flow since this information could be extrapolated to most of the volcanoes of the southern Andes, all of which are exposed to high rainfall.

Back-analysis conclude that the 2017 event occurred in a small area. In-situ observations allowed to establish that possible liberation zones could generate greater debris flow with similar water contents. Our projections indicate that the final volume generated could vary between 32,039 and 688,142 m³ depending of the initial water content. Small volumes projected

suggest that it could be generated continuously without reach populated areas. Constant pluvial and fluvial erosion in the study zone could cause larger rockfalls and slips, causing more catastrophic events in the future. Therefore, we propose that the Osorno volcano need to be taken into account like a hot-spot for debris-flow monitoring.

Finally, our results report for the first time a case of a debris flow on a stratovolcano in the southern Andes from its generation, using data collection in situ and modelling with sensitivity analysis using a two-phase model. Our conclusion is

to urge the scientific community to focus efforts on generating scenarios for debris flows in the stratovolcanoes of the southern Andes. The population density around Osorno volcano is low but receives a high number of tourists every year However, the areas of stratovolcanoes like Villarrica and Calbuco contain higher population densities, and these volcanoes must not be ignored in future territorial plans. The present study shows evidence that the debris flows identified are recurrent events, even though they do not always reach populated areas. Our results show that this threat is inherent to volcanic

activity, so any future risk analysis must consider debris flows. This will allow better risk management in nearby population centers and a safer coexistence with the volcanic structures of the southern Andes.

Author contributions

IF and BM contributed to the conceptualization and methodology of the research and performed the formal analysis, visualization, and validation. IF and MS was involved in the funding and supervision of the paper. IF and PM contributed

with the supervision, review, and editing of the paper. RM, IR and NC provided input in terms of methodology and the review and editing of the paper.





## Competing interests

The authors declare that they have no conflict of interest.

## Financial support

This was made possible thanks to the "Agencia Nacional de Investigación y Desarrollo (ANID)" of the Chilean government, grant number PII180008 and the "Fondecyt Iniciación" program (grant no. 11180500).

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
