# Peer review of "Debris Flow event on Osorno volcano, Chile, during summer 2017: New interpretations for chain processes in the Southern Andes."

_Natural Hazards and Earth System Sciences, 2021_

## Author Comment (AC1)

**Debris Flow event on Osorno volcano, Chile, during summer 2017: New interpretations for chain processes in the Southern Andes.**

Ivo Janos Fustos-Toribio[1], Bastian Morales-Vargas[2,3], Marcelo Somos-Valenzuela[3,4*], Pablo Moreno-Yaeger [1,5], Ramiro Muñoz-Ramirez[2], Ines Rodriguez Araneda[2], Ningsheng Chen[6]

[1] Department of Civil Engineering, University of La Frontera, Francisco Salazar 1145, Temuco, Chile
[2] Departamento de Obras Civiles y Geología, Facultad de Ingeniería, Universidad Católica de Temuco, Rudecindo Ortega 02950, Temuco, Chile
[3] Department of Forest Sciences, Faculty of Agriculture and Forest Sciences, Universidad de La Frontera, Av. Francisco Salazar 01145, Temuco, Chile, 4780000
[4] Butamallín Research Center for Global Change, University of La Frontera, Av. Francisco Salazar 01145, Temuco, Chile, 4780000
[5] Department of Geoscience, University of Wisconsin–Madison, 1215 West Dayton St., Madison, WI 53706, USA.
[6] Key Laboratory of Mountain Hazards and Surface Processes, Institute of Mountain Hazards and Environment, Chinese Academy of Sciences, Chengdu 610041, China

1. Reviewer 1: Abstract

   The abstract is comprehensive and appropriate for the paper.

   *I suggest to remove the word "Finally" at the end of the abstract.

   > A: We removed "Finally"

Introduction

I think the description of the volcanic flow-type landslides in the Andes is very interesting. Probably mentions of the numerical modeling of debris flows in the scientific literature should be included.

2. Reviewer 1: Line 38. "Nevertheless, debris flows in volcanic zones have not been evaluated in detail". I think this phrase is too risky. You can only say that it requires more investigation or not many research studies have been carried out…

   A: We modified "Nevertheless, debris flows in volcanic zones have not been evaluated in detail... " to "Nevertheless, debris flows in volcanic zones have not many research studies have been carried out". Thanks by your accotation.

3. Reviewer 1: Figure 1. It is difficult to read the text of the figure. Maybe the font size is too small or probably it is solved easily increasing the size of the figure.

   A: We change figure 1 with an improved figure with higher resolution.

[Figure]

**Area and points of interest**

- ⬚ Proximal zone
- ⬚ Distal zone
- 🔵 Liberation calibration
- 🔴 Liberation zone 1
- 🟢 Liberation zone 2
- 🟡 Liberation zone 3
- 🟠 Liberation zone 4
- 🟢 Geotechnical probe
- ⬟ Figure 3
- ⬟ Figure 4
- ⬟ Figure 8
- 🔵 Petrohué
- ━━ Route CH-255

**Sedimentary deposits and volcanoclastic**

**Holocene**
- Sedimentary and volcanoclastic deposits
- Morrenic deposits recent (< 1600 AD)

**Upper Pleistocene**
- Morrenic deposits tardiglacial (32-14 ka)
- Morrenic deposits old (> 32 ka)

**Osorno volcano**

**Holocene**
- Unit Osorno 3-4 Lavas basaltic and andesitc-basaltic (12-<4 ka)

**Upper Pleistocene**
- Unit Osorno 1-2 Volcano-sedimentary rocks with intercalation of basaltic rocks (200-110 ka) and Lavas basaltic with intercalation piroclastic (90-12 ka)

**Basement**
- Basement (70-18 Ma)

**Other volcanoes**
- La Picada (500-60 ka)

**Adventitious eruptive centers**
- Pyroclastic cones, fissures eruptive, dacitic domes and associated lavas (11 ka-1835 AD)

[Figure]

**Area and points of interest**

- ⬭ Proximal zone
- ⬭ Distal zone
- 🔵 Liberation calibration
- 🔴 Liberation zone 1
- 🟢 Liberation zone 2
- 🟡 Liberation zone 3
- 🔴 Liberation zone 4
- 🟢 Geotechnical probe
- ⬟ Figure 3
- 🟡 Figure 4
- ⬟ Figure 8
- 🔵 Petrohué
- 🟣 Solitario test point

**Holocene**

- Sedimentary, volcanoclastic and morrenic deposits (< 1600 AD)

- Unit Osorno 3-4
  Lavas basaltic and andesitc-basaltic (12-<4 ka)

**Upper Pleistocene**

- Morrenic deposits tardiglacial (> 14 ka)

- Unit Osorno 1-2
  Volcano-sedimentary rocks with intercalation of basaltic rocks (200-110 ka) and Lavas basaltic with intercalation piroclastic (90-12 ka)

**Basement**

- Basement (70-18 Ma)

**Other volcanoes**

- La Picada (500-60 ka)

**Adventitious eruptive centers**

- Pyroclastic cones, fissures eruptive, dacitic domes and associated lavas (11 ka-1835 AD)

4. Reviewer 1: Line 114. I suggest to mention Figure 2 better (in the text).

   A: Now we added

   "We implement a methodological approach based on comprehensive numerical modelling constrained by field data and laboratory analysis"

5. Reviewer 1: "Different soil deposits allowed understood the mobilization features, checking if it corresponds to a non-Newtonian flow." I suggest to rewrite this sentence.

    A: Now we rewrite the sentence from

    Different soil deposits allowed understood the mobilization features, checking if it corresponds to a non-Newtonian flow.

    to:

    Moreover, debris flow deposits identified in the field allowed understanding the rheology of these events (non-Newtonian flows).

6. Reviewer 1: "Liberation zones close to the Volcano summit, were physical weathering of the lava flow deposits exposed by pluvial erosion was assessed in the field." And to check (and correct) this one. "in the follow numerical model."

    A: We modified the following text:

    "Liberation zones close to the Volcano summit, were physical weathering of the lava flow deposits exposed by pluvial erosion was assessed in the field." to "We evaluated debris flow initiation zones close to the Volcano summit and the physical weathering of rock/soil."

    "Scarps with potential rockfalls of unstable blocks were identified, measured, and georeferenced to stablish these zones in the follow numerical model." to "Scarps with potential rockfalls of unstable blocks were identified, measured, and georeferenced. We established these scarps as initiation debris flow zones in the following numerical model."

7. Reviewer 1: "The estimated values of the cohesion and internal friction angle was integrated into a database to be used during the back analysis phase" ...were integrated

    A: Corrected

8. Reviewer 1: Line 153. The first time you mention "SERNAGEOMIN" you should specify what it is.

    A: We change the sentence from

    The back-analysis considered a final height of 1.5 m in route CH255, according to reports of SERNAGEOMIN.

    To

The back-analysis considered a final height of 1.5 m in route CH255, according to reports of National Geological and Mining Survey (SERNAGEOMIN in Spanish).

9.  Reviewer 1: Line 160. "If we take the ?f correspond to the fluid viscosity with isotropic stress distribution." Please rewrite it clearer.

    A: Corrected, thanks by your detailed comment.

10. Reviewer 1: "Table 2Model parameters usd in r.avaflow." A space is missing. Correct the word "used." Why is there an asterisk (*) in the internal friction angle?

    A: Now we added a space and replaced "usd" by "used"

11. Reviewer 1: Line 168. "parameters presenting serious discrepancies were..." What do you mean serious discrepancies? Is it discrepancies between the simulations and the real event? Or measurements?

    A: We modified "In a first approach, parameters presenting serious discrepancies were restricted to the range proposed by Pudasaini (2012) for the simulation of debris flows." to "We selected model parameters to the range proposed by Pudasaini (2012) for the simulation of debris flows. We compared height flow with the calibration point (route CH-255), discarding unreasonable simulations."

12. Reviewer 1:  Line 170. Do you consider a 30-m spatial resolution is appropriate for the modeling? Now, I think that if the reason for the selection of the spatial resolution is the availability or considerable computing efforts it should be stated in the paper. Line 184. "Surface representation" Good, partially answering my previous questions.

    A: Now, we added an additional statement. We modified from "The SRTM and ASTER-GDEM models were used separately, with a spatial resolution of 30 meters" to "The SRTM and ASTER-GDEM models were used separately, with a spatial resolution of 30 meters due to data availability limitations."

13. Reviewer 1: Line 205. "These which favours a faster movement, increasing lateral erosion according to field results" Please rewrite if it can be clearer.

    A: We modify from "These which favours a faster movement, increasing lateral erosion according to field results [Shu et al., 2018]." to "Lower unit weight favours faster flow processes increasing lateral erosion [Shu et al., 2018] and concordant with lateral erosion observed."

14. Reviewer 1: Figure 4 is referenced before Figure 3.

A: We inverted the figures to be coherent within the text

15. Reviewer 1:  Line 233. "allowed understood" Probably mispelled.

A: We modified "Different initial water content into the simulations and DEMs differences allowed understood the uncertainty of the main initial input." to "Different initial water content into the simulations and DEMs differences delimited the uncertainty of the main initial input."

16. Reviewer 1: Line 234. "(¡Error! No se encuentra el origen de la referencia)"

A: Removed

17. Reviewer 1: Figure 5. "Volumen" Volume.

A: We corrected "Volumen" to "Volume" in figure 5 and 7.

18. Reviewer 1: Figure 6. "Escarp". "Error backanalysis". Scarp. And Reviewer 2: Figure 6: It would be useful to report the boundaries of the real debris flow for comparison with the simulations.

A: We corrected figure 6. Thanks for your detailed observation and corrections.

19. Reviewer 1: I think the results presented are relevant to the scientific community. I wonder if the authors consider that figures including more modeling results were not so relevant. In my opinion, since different scenarios, projections and zones were modeled, more maps of the debris flows would be interesting. Probably to include it (I suppose the journal have this option) as supplementary material if the authors justify that they were not so relevant for the paper.

A: We select the simulations that had relevant scientific information in the supplementary material in the original submission. If the reviewer needs additional information, we will add them.

20. Reviewer 1:  I'm not sure if it was specified how the error calculation was done.

A: We modified from "Hence, we calibrate the flow runout taking control points of the height of the flow measured on the main road minutes after the event. We also assessed the quality of the simulations using possible release volumes based on field evidence." to "Hence, we calibrate the flow runout taking control points of the height of the flow measured on the main road minutes after the event. The percentual error was calculated using the height simulated with the measured height in "El Solitario pass" (Figure 1). A percentual difference was used between the simulated value and the measured height divided by the measured height. We also assessed the quality of the simulations using possible release volumes based on field evidence."

21. Reviewer 1: "DEM SRTM (...) DEM ASTER concentrated a larger part of the material towards the north". Not clear for me.

> A: We modified "DEM SRTM produced a variation in the flow to the south, increasing the impact in Route CH-255; in contrast, DEM ASTER concentrated a larger part of the material towards the north" to "Simulations using DEM SRTM produced a variation of the flow to the south, increasing the impact in Route CH-255. These results had differences with the simulations using DEM ASTER, which showed the flow towards the north."

22. Reviewer 1: I consider many figures (e.g. 5-7) should be referenced better if divided to Figure 7a, b, c...

> A: Figures 5 and 7 represent different phases of our contribution, so merge in just one could be confusing. We tried to merge figures 5 and 7 into just one, becoming too confusing to interpret. We request that these figures will be considered independent pictures in the final manuscript.

23. Reviewer 1: Lines 315-320. It is good for my previous comment about the DEM resolutions.

> A: We agree with the previous comment of the reviewer. The zone has not Lidar data available or DEM with very high spatial resolution. Therefore, we discussed the results of two surface products available.

24. Reviewer 1: "Finally, water-rich mass flows are distinguished by material type, water content, the presence of excess pore pressure, or liquefaction at the source (Calhoun & Clague, 2018)." This phrase is repeated in lines 337 and 349. Then, both paragraphs have the same contents. Probably you were supposed to remove one of them.

> A: Our apologies, we removed the second paragraphs. Thank you for the detailed comment.

25. Reviewer 1: Not clear about "best modelled scenario" and "worse scenario for...".

> A: We modified "On the other hand, the area affected by the removal is greater with SRTM, for which we propose that the best modelled scenario represents at the same time the worst scenario for the resident population in the study area." to "Moreover, the debris flows area is greater using the DEM SRTM in comparison to DEM ASTER. The analysis using two DEMs allows evaluating the debris flow susceptibility of the population in the study area."

26. Reviewer 1: I suggest to shorten the first three paragraphs of the conclusions. A summary may be acceptably but I consider it is not necessary to present results to conclude about them (at least taking so many long phrases). The last conclusion is very accurate.

A: We modified the conclusions to avoid repeat results and previous information.

---

## Author Comment (AC2)

**Debris Flow event on Osorno volcano, Chile, during summer 2017: New interpretations for chain processes in the Southern Andes.**

Ivo Janos Fustos-Toribio[1], Bastian Morales-Vargas[2,3], Marcelo Somos-Valenzuela[3,4*], Pablo Moreno-Yaeger [1,5], Ramiro Muñoz-Ramirez[2], Ines Rodriguez Araneda[2], Ningsheng Chen[6]

[1] Department of Civil Engineering, University of La Frontera, Francisco Salazar 1145, Temuco, Chile
[2] Departamento de Obras Civiles y Geología, Facultad de Ingeniería, Universidad Católica de Temuco, Rudecindo Ortega 02950, Temuco, Chile
[3] Department of Forest Sciences, Faculty of Agriculture and Forest Sciences, Universidad de La Frontera, Av. Francisco Salazar 01145, Temuco, Chile, 4780000
[4] Butamallín Research Center for Global Change, University of La Frontera, Av. Francisco Salazar 01145, Temuco, Chile, 4780000
[5] Department of Geoscience, University of Wisconsin–Madison, 1215 West Dayton St., Madison, WI 53706, USA.
[6] Key Laboratory of Mountain Hazards and Surface Processes, Institute of Mountain Hazards and Environment, Chinese Academy of Sciences, Chengdu 610041, China

1. Reviewer 2: Equation 1: Please, define the symbol "A" and the underscore symbol.

   A: Now we added the description

   Insert text: "… distribution and $A(\alpha_f)$ is called the mobility of the fluid at the interface."

2. Reviewer 2: Table 1 reports parameters with different units respect to those used elsewere in the text. I suggest the following corrections: Natural unit weight -> Natural density, Dry unit weight -> Density, Unit weight after consolidation -> Density after consolidation.

   A: Modified

3. Reviewer 2: Concerning the last to lines of Table 1, "shear force" and "normal force" are actually "shear stress" and "normal stress". Moreover, I suggest to adopt SI units and convert the stress unit from kg/cm2 to N/m2 or N/cm2. Table 2 reports the density in kg/m3, differently from Table 1, which reports densities in g/cm3. This is not an error, but it would be better to adopt the same units across the paper (See previous comment on Table 1). I propose to mantain the units adopted in Table 2 and modify those of Table 1.

   A: We modified shear and normal forces to shear and normal stress. Moreover, we used same units for densities to be consistent with table 2. Finally, the stress unit are on N/m2

4. Reviewer 2: Moreover, in Table 2, the decimal separator (dot) is intermixed with the decimal separator (comma). I suggest to uniform the separators by using the dot for separating the decimals.

   A: We correct this typo using only dots.

5. Reviewer 2: Header of Table 2: usd -> used

   A: Corrected

6. Reviewer 2: Line 78: "routethe" -> "route the"

   A: Corrected

7. Reviewer 2: Line 125: cannel -> channel

   A: Corrected

8. Reviewer 2: Lines 234-235: It seems that a reference cannot be found

   A: Removed

9. Reviewer 2: Line  342: 105 -> 10^5

   A: Corrected